# MDocAgent: A Multi-Modal Multi-Agent Framework for Document Understanding

## Abstract

Document Question Answering (DocQA) is a very common task. Existing methods using Large Language Models (LLMs) or Large Vision Language Models (LVLMs) and Retrieval Augmented Generation (RAG) often prioritize information from a single modal, failing to effectively integrate textual and visual cues. These approaches struggle with complex multi-modal reasoning, limiting their performance on real-world documents. We present MDocAgent (A Multi-Modal Multi-Agent Framework for Document Understanding), a novel RAG and multi-agent framework that leverages both text and image. Our system employs five specialized agents: a general agent, a critical agent, a text agent, an image agent and a summarizing agent. These agents engage in multi-modal context retrieval, combining their individual insights to achieve a more comprehensive understanding of the document's content. This collaborative approach enables the system to synthesize information from both textual and visual components, leading to improved accuracy in question answering. Preliminary experiments on five benchmarks like MMLongBench, LongDocURL demonstrate the effectiveness of our MDocAgent, achieve an average improvement of 12.1% compared to current state-of-the-art method. This work contributes to the development of more robust and comprehensive DocQA systems capable of handling the complexities of real-world documents containing rich textual and visual information.

## 1 Introduction

Answering questions based on reference documents (DocQA) is a critical task in many applications (Ding et al., 2022; Tanaka et al., 2023; Mishra et al., 2019; Cho et al., 2024; Zhang et al., 2024a; Ma et al., 2024a; Suri et al., 2024), ranging from information retrieval to automated document analysis. A key challenge in DocQA lies in the diverse nature of questions and the information needed to answer them (Ma et al., 2024b; Deng et al., 2024). Questions can refer to textual content, to visual elements within the document (e.g., charts, diagrams, images), or even require the integration of information from both modalities. Since Large Language Models (LLMs) can only handle textual information (Naveed et al., 2023), Large Vision Language Models (LVLMs) are often used in DocQA (Luo et al., 2024; Hu et al., 2024; Chen et al., 2024). As illustrated in Figure 1, while LVLMs have shown promise in handling visual content, they often struggle in scenarios where key information is primarily textual, or where a nuanced understanding of the interplay between text and visual elements is required (Cho et al., 2024; Ma et al., 2024a; Suri et al., 2024). Another challenge in DocQA lies in the huge volume of information often present in documents. Processing entire documents directly can overwhelm computational resources and make it difficult for models to identify the most pertinent information (Ma et al., 2024b; Deng et al., 2024).

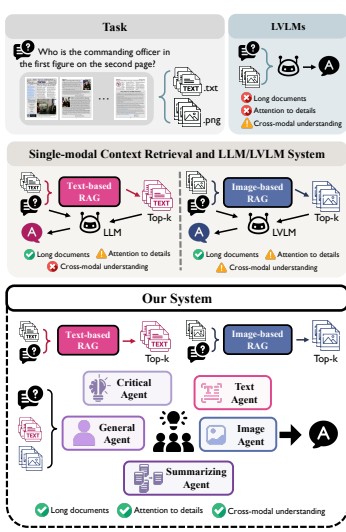

Figure 1: Comparison of different approaches for DocQA.

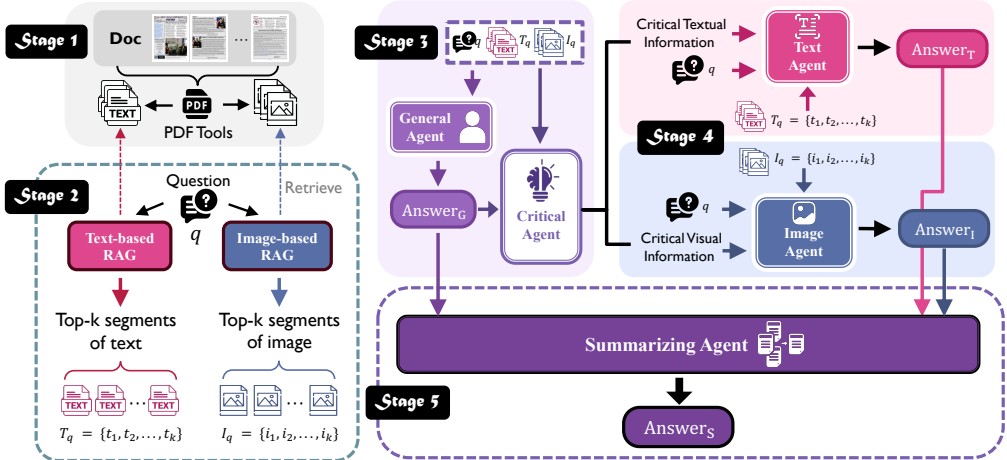

Figure 2: Overview of **MDocAgent**: A multi-modal multi-agent framework operating in five stages: (1) Documents are processed using PDF tools to extract text and images. (2) Text-based and image-based RAG retrieves the top-k relevant segments and image pages. (3) The general agent provides a preliminary answer, and the critical agent extracts critical information from both modalities. (4) Specialized agents process the retrieved information and critical information within their respective modalities and generate refined answers. (5) The summarizing agent integrates all previous outputs to generate the final answer.

To overcome this challenge, Retrieval Augmented Generation (RAG) is used as an auxiliary tool to extract the critical information from a long document (Gao et al., 2023). While RAG methods like ColBERT (Khattab & Zaharia, 2020) and ColPali (Faysse et al., 2024a) have proven effective for retrieving textual or visual information respectively, they often fall short when a question requires integrating insights from both modalities. Existing RAG implementations typically operate in isolation, either retrieving text or images (Lewis et al., 2020; Xia et al., 2024c), but lack the ability to synthesize information across these modalities. Consider a document containing a crucial diagram and accompanying textual explanations. If a question focuses on the diagram's content, a purely text-based RAG system would struggle to pinpoint the relevant information. Conversely, if the question pertains to a nuanced detail within the textual description, an image-based RAG would be unable to isolate the necessary textual segment. This inability to effectively combine multi-modal information restricts the performance of current RAG-based approaches in complex DocQA tasks. Moreover, the diverse and nuanced nature of these multimodal relationships requires not just retrieval, but also a mechanism for reasoning and drawing inferences across different modalities.

To further address these limitations, we present a novel framework, a Multi-Modal Multi-Agent Framework for Document Understanding (MDocAgent), which leverages the power of both RAG and a collaborative multi-agent system where specialized agents collaborate to process and integrate text and image information. MDocAgent employs two parallel RAG pipelines: a text-based RAG and an image-based RAG. These retrievers provide targeted textual and visual context for our multi-agent system. MDocAgent comprises **five specialized agents**: a general agent for initial multi-modal processing, a critical agent for identifying key information, a text agent, an image agent for focused analysis within their respective modalities, and a summarizing agent to synthesize the final answer. This collaborative approach enables our system to effectively tackle questions that require synthesizing information from both textual and visual elements, going beyond the capabilities of traditional RAG methods.

Specifically, MDocAgent operates in five stages: **(1) Document Pre-processing:** Text is extracted via OCR and pages are preserved as images. **(2) Multi-modal Context Retrieval:** text-based and image-based RAG tools retrieve the top-k relevant text segments and image pages, respectively. **(3) Initial Analysis and Key Extraction:** The general agent generates an initial response, and the critical agent extracts key information, providing it to the specialized agents. **(4) Specialized Agent Processing:** Text and image agents analyze the retrieved context within their respective modalities, guided by the critical information. **(5) Answer Synthesis:** The summarizing agent integrates all agent responses to produce the final answer.

The primary contribution of this paper is a novel multi-agent framework for DocQA that effectively integrates specialized agents, each dedicated to a specific modality or aspect of reasoning, including text and image understanding, critical information extraction, and answer synthesis. We demonstrate the efficacy of our approach through experiments on five benchmarks: MMLongBench (Ma et al., 2024b), LongDocURL (Deng et al., 2024), PaperTab (Hui et al., 2024), PaperText (Hui et al., 2024), and FetaTab (Hui et al., 2024), showing significant improvements in DocQA performance, with an average of 12.1% compared to current SOTA method. The empirical improvements demonstrate the effectiveness of our collaborative multi-agent architecture in handling long, complex documents and questions. Furthermore, ablation studies validate the contribution of each agent and the importance of integrating multi-modalities.

## 2 RELATED WORK

**LVLMs in DocQA Tasks.** Document Visual Question Answering (DocVQA) has evolved from focusing on short documents to handling complex, long, and multi-document tasks (Ding et al., 2022; Tanaka et al., 2023; Mishra et al., 2019; Tito et al., 2023), often involving visually rich content such as charts and tables. This shift requires models capable of integrating both textual and visual information. Large Vision Language Models (LVLMs) have emerged to address these challenges by combining the deep semantic understanding of Large Language Models (LLMs) with the ability to process document images (Liu et al., 2024a;b; Zhu et al., 2023; Dai et al., 2023; Zhou et al., 2023; 2024a;b; Xia et al., 2024a;b; Zhu et al., 2024; Zhang et al., 2024b; Tong et al., 2025). LVLMs convert text in images into visual representations, preserving layout and visual context. However, they face challenges like input size limitations and potential loss of fine-grained textual details (Luo et al., 2024; Hu et al., 2024), making effective integration of text and visual information crucial for accurate DocVQA performance (Park et al., 2024).

**Retrieval-Augmented Generation.** Retrieval Augmented Generation (RAG) enhances LLMs by supplying them with external text-based context, thereby improving their performance in tasks such as DocQA (Lewis et al., 2020; Gao et al., 2023). Recently, with the increasing prevalence of visually rich documents, image RAG approaches have been developed to retrieve relevant visual content for Large Vision Language Models (LVLMs) (Xia et al., 2024d;c; Cho et al., 2024; Chen et al., 2024; Xing et al., 2025). However, existing methods struggle to effectively integrate and reason over both text and image information, as retrieval often occurs independently. This lack of integrated reasoning limits the effectiveness of current RAG techniques, especially for complex DocQA tasks that require a nuanced understanding of both modalities.

**Multi-Agent Systems.** Multi-agent systems have shown promise in complex domains like medicine (Wu et al., 2023; Li et al., 2023; Kim et al., 2024). These systems use specialized agents to focus on different task aspects (Chan et al., 2023; Kannan et al., 2024; Li et al., 2025), collaborating to achieve goals that a single model may struggle with. However, their application to DocQA introduces unique challenges stemming from the need to integrate diverse modalities. Simply combining the outputs of independent text and image agents often fails to capture the nuanced interplay between these modalities, which is crucial for accurate document understanding. Our framework addresses this by introducing *a general agent for information integration* alongside specialized text and image agents, enabling collaborative reasoning and a more comprehensive understanding of document content, ultimately improving DocVQA performance.

## 3 MULTI-MODAL MULTI-AGENT FRAMEWORK FOR DOCUMENT UNDERSTANDING

This section details our proposed framework, MDocAgent, for tackling the complex challenges of DocQA. MDocAgent employs a novel five-stage multi-modal, multi-agent approach as shown in Figure 2, utilizing specialized agents for targeted information extraction and cross-modal synthesis to achieve a more comprehensive understanding of document content. Subsequently, Section 3.1 through Section 3.5 provide a comprehensive description of MDocAgent's architecture. This detailed exposition will elucidate the mechanisms by which MDocAgent effectively integrates and leverages textual and visual information to achieve improved accuracy in DocQA.

**Preliminary: Document Question Answering.** Given a question $q$ expressed in natural language and the corresponding document $\mathcal{D}$, the goal is to generate an answer a that accurately and comprehensively addresses $q$ using the information provided within $\mathcal{D}$.

## 3.1 DOCUMENT PRE-PROCESSING

This initial stage prepares the document corpus for subsequent processing by transforming it into a format suitable for both textual and visual analysis. $\mathcal{D}$ consists of a set of pages $\mathcal{D} = \{p_1, p_2, \ldots, p_N\}$. For each page $p_i$, textual content is extracted using a combination of Optical Character Recognition (OCR) and PDF parsing techniques. OCR is employed to recognize text within image-based PDFs, while PDF parsing extracts text directly from digitally encoded text within the PDF. This dual approach ensures robust text extraction across various document formats and structures. The extracted text for each page $p_i$ is represented as a sequence of textual segments or paragraphs $t_i = \{t_{i1}, t_{i2}, \ldots, t_{iM}\}$, where $M$ represents the number of text segments on that page. Concurrently, each page $p_i$ is also preserved as an image, retaining its original visual layout and features. This allows the framework to leverage both textual and visual cues for comprehensive understanding. This pre-processing results in two parallel representations of the document corpus: a textual representation consisting of extracted text segments and a visual representation consisting of the original page images. This dual representation forms the foundation for the multi-modal analysis performed by the framework.

## 3.2 MULTI-MODAL CONTEXT RETRIEVAL

The second stage focuses on efficiently retrieving the most relevant information from the document corpus, considering both text and image modalities. Algorithm 1 illustrates the whole procedure of retrieval. For the textual retrieval, extracted text segments $t_i$ of each page $p_i$ are indexed using ColBERT (Khattab & Zaharia, 2020). Given the user question $q$, ColBERT retrieves the top-$k$ most relevant text segments, denoted as $T_q = \{t_1, t_2, \ldots, t_k\}$. This provides the textual context for subsequent agent processing. Parallel to textual retrieval, visual context is extracted using ColPali (Faysse et al., 2024a). Each page image $p_i$ is processed by ColPali to generate a dense visual embedding $E^{p_i} \in \mathbb{R}^{n^v \times d}$, where $n^v$ represents the number of visual tokens per page and $d$ represents the embedding dimension. Using these embeddings and the question $q$, ColPali retrieves the top-$k$ most visually relevant pages, denoted as $I_q = \{i_1, i_2, \ldots, i_k\}$. The use of ColPali allows the model to capture the visual information present in the document, including layout, figures, and other visual cues.

---

**Algorithm 1** Multi-modal Context Retrieval

---

**Require:** Question $q$, Document $D$, Text Scores $S_t$, Image Scores $S_i$, Text Relevance Scores $R_t$, Image Relevance Scores $R_i$.
**Ensure:** Top-k text segments $T_q$, Top-k image segments $T_q$.
 1: $S_t \leftarrow \{\}$
 2: $S_i \leftarrow \{\}$                                     ▷ Iterate through each page in the corpus
 3: **for** each $p$ in $D$ **do**
 4:     **for** each text segment $t$ in $p$ **do**
 5:         $S_t[t] \leftarrow R_t(q, t)$                        ▷ Calculate text relevance score
 6:     **end for**
 7:     $S_i[p] \leftarrow R_i(q, p)$                       ▷ Calculate image relevance score
 8: **end for**
 9: $T_q \leftarrow \text{Top\_K}(S_t, k)$                   ▷ Select top-k text segments
10: $I_q \leftarrow \text{Top\_K}(S_i, k)$                  ▷ Select top-k image segments
11: **return** $T_q, I_q$

---

## 3.3 INITIAL ANALYSIS AND KEY EXTRACTION

The third stage aims to provide an initial interpretation of the question and pinpoint the most salient information within the retrieved context. The general agent $A_G$, functioning as a preliminary multi-modal integrator, receives both the retrieved textual context $T_q$ and the visual context $I_q$. It processes these multimodal inputs by effectively combining the information embedded within both modalities.

This comprehensive understanding of the combined context allows $A_G$ to generate a preliminary answer $a_G$, which serves as a crucial starting point for more specialized analysis in the next stage.

$$a_G = A_G(q, T_q, I_q). \tag{1}$$

Subsequently, the critical agent $A_C$ plays a vital role in refining the retrieved information. It takes as input the question $q$, the retrieved contexts $T_q$ and $I_q$, and the preliminary answer $a_G$ generated by the general agent. The primary function of $A_C$ is to meticulously analyze these inputs and identify the most crucial pieces of information that are essential to accurately answer the question. This critical information acts as a guide for the specialized agents in the next stage, focusing their attention on the most relevant aspects of the retrieved context.

$$T_c = A_C(q, T_q, a_G), \quad I_c = A_C(q, I_q, a_G). \tag{2}$$

The output of this stage consists of $T_c \subset T_q$, representing the critical textual information extracted from the retrieved text segments, and $I_c$, which provides a detailed textual description of the critical visual information extracted from the retrieved images $I_q$ that capture the essence of the important visual elements.

### 3.4 SPECIALIZED AGENT PROCESSING

The fourth stage delves deeper into the textual and visual modalities, leveraging specialized agents guided by the critical information extracted in the previous stage. The text agent $A_T$ receives the retrieved text segments $T_q$ and the critical textual information $T_c$ as input. It operates exclusively within the textual domain, leveraging its specialized knowledge and analytical capabilities to thoroughly examine the provided text segments. By focusing specifically on the critical textual information $T_c$, $A_T$ can pinpoint the most relevant evidence within the broader textual context $T_q$ and perform a more focused analysis. This focused approach allows for a deeper understanding of the textual nuances related to the question and culminates in the generation of a detailed, text-based answer $a_T$.

$$a_T = A_T(q, T_q, T_c). \tag{3}$$

Concurrently, the image agent $A_I$ receives the retrieved images $I_q$ and the critical visual information $I_c$. This agent specializes in visual analysis and interpretation. It processes the images in $I_q$, paying particular attention to the regions or features highlighted by the critical visual information $I_c$. This targeted analysis allows the agent to extract valuable insights from the visual content, focusing its processing on the most relevant aspects of the images. The image agent's analysis results in a visually-grounded answer $a_I$, which provides a response based on the interpretation of the images.

$$a_I = A_I(q, I_q, I_c). \tag{4}$$

### 3.5 ANSWER SYNTHESIS

The final stage integrates the diverse outputs from the preceding stages, combining the initial multimodal understanding with the specialized agent analyses to produce a comprehensive and accurate answer. The summarizing agent $A_S$ receives the answers $a_G$, $a_T$, and $a_I$ generated by the general agent, text agent, and image agent, respectively. This comprehensive set of information provides a multifaceted perspective on the question and allows the summarizing agent to perform a thorough synthesis. The summarizing agent analyzes the individual agent answers, identifying commonalities, discrepancies, and complementary insights. It considers the supporting evidence provided by each agent. By resolving potential conflicts or disagreements between the agents and integrating their individual strengths, the summarizing agent constructs a final answer $a_S$ that leverages the collective intelligence of the multi-agent system. This final answer is not merely a combination of individual answers but a synthesized response that reflects a deeper and more nuanced understanding of the information extracted from both textual and visual modalities. The whole procedure of this multi-agent collaboration is illustrated in Algorithm 2.

## 4 EXPERIMENTS

We evaluate MDocAgent on five document understanding benchmarks covering multiple scenarios to answer the following questions: (1) Does MDocAgent effectively improve document understanding accuracy compared to existing RAG-based approaches? (2) Does each agent in our framework play a meaningful role? (3) How does our approach enhance the model's understanding of documents?

---

**Algorithm 2** Multi-agent Collaboration

---

**Require:** Question $q$, Top-k text segments $T_q$, Top-k image segments $I_q$, General Agent $A_G$, Critical Agent $A_C$, Text Agent $A_T$, Image Agent $A_I$, Summarizing Agent $A_S$
**Ensure:** Final answer $a_s$,
 1: $a_G \leftarrow A_G(q, T_q, I_q)$               ▷ General agent answer
 2: $(T_c, B_c) \leftarrow A_C(q, T_q, I_q, a_G)$         ▷ Extract critical info
 3: $a_T \leftarrow A_T(q, T_q, T_c)$                ▷ Text agent answer
 4: $a_I \leftarrow A_I(q, I_q, B_c)$                ▷ Image agent answer
 5: $a_S \leftarrow A_S(q, a_G, a_T, a_I)$         ▷ Final answer synthesis
 6: **return** $a_S$

---

Table 1: Performance comparison across MDocAgent and existing state-of-the-art LVLMs and RAG-based methods.

| Method | MMLongBench | LongDocUrl | PaperTab | PaperText | FetaTab | Avg |
|---|---|---|---|---|---|---|
| *LVLMs* | | | | | | |
| Qwen2-VL-7B-Instruct | 0.165 | 0.296 | 0.087 | 0.166 | 0.324 | 0.208 |
| Qwen2.5-VL-7B-Instruct | 0.224 | 0.389 | 0.127 | 0.271 | 0.329 | 0.268 |
| LLaVA-v1.6-Mistral-7B | 0.099 | 0.074 | 0.033 | 0.033 | 0.110 | 0.070 |
| Phi-3.5-Vision-Instruct | 0.144 | 0.280 | 0.071 | 0.165 | 0.237 | 0.179 |
| LLaVA-One-Vision-7B | 0.053 | 0.126 | 0.056 | 0.108 | 0.077 | 0.084 |
| SmolVLM-Instruct | 0.081 | 0.163 | 0.066 | 0.137 | 0.142 | 0.118 |
| *RAG methods (top 1)* | | | | | | |
| ColBERTv2+LLaMA-3.1-8B | 0.241 | 0.429 | 0.155 | 0.332 | 0.490 | 0.329 |
| M3DocRAG (ColPali+Qwen2-VL-7B) | 0.276 | 0.506 | 0.196 | 0.342 | 0.497 | 0.363 |
| **MDocAgent (Ours)** | **0.299** | **0.517** | **0.219** | **0.399** | **0.600** | **0.407** |
| *RAG methods (top 4)* | | | | | | |
| ColBERTv2+LLaMA-3.1-8B | 0.273 | 0.491 | 0.277 | 0.460 | 0.673 | 0.435 |
| M3DocRAG (ColPali+Qwen2-VL-7B) | 0.296 | 0.554 | 0.237 | 0.430 | 0.578 | 0.419 |
| **MDocAgent (Ours)** | **0.315** | **0.578** | **0.278** | **0.487** | **0.675** | **0.465** |

## 4.1 EXPERIMENT SETUP

**Implementation Details**. There are five agents in MDocAgent: general agent, critical agent, text agent, image agent and summarizing agent. We adopt Llama-3.1-8B-Instruct (Grattafiori et al., 2024) as the base model for text agent, Qwen2-VL-7B-Instruct (Wang et al., 2024) for other four agents, and select ColBERTv2 (Santhanam et al., 2021) and ColPali (Faysse et al., 2024b) as the text and image retrievers, respectively. In our settings of RAG, we retrieve 1 or 4 highest-scored segments as input context for each example. All experiments are conducted on 4 NVIDIA H100 GPUs. Details of models and settings are shown in Appendix A.

**Datasets**. The benchmarks involve MMLongBench (Ma et al., 2024b), LongDocUrl (Deng et al., 2024), PaperTab (Hui et al., 2024), PaperText (Hui et al., 2024), FetaTab (Hui et al., 2024). These evaluation datasets cover a variety of scenarios, including both open- and closed-domain, textual and visual, long and short documents, ensuring fairness and completeness in the evaluation. Details of dataset descriptions are in Appendix A.2.

**Metrics**. For all benchmarks, following (Ma et al., 2024b; Deng et al., 2024), we leverage GPT-4o (OpenAI, 2023) as the evaluation model to assess the consistency between the model's output and the reference answer, producing a binary decision (correct/incorrect). We provide the average accuracy rate for each benchmark.

## 4.2 MAIN RESULTS

In this section, we provide a comprehensive comparison of MDocAgent on multiple benchmarks against existing state-of-the-art LVLMs and RAG-based methods built on them. Our findings can be summarized as:

Table 2: Performance comparison across different MDocAgent's variants.

| Variants | Agent Configuration | | | Evaluation Benchmarks | | | | | |
|---|---|---|---|---|---|---|---|---|---|
| | General & Critical Agent | Text Agent | Image Agent | MMLongBench | LongDocUrl | PaperTab | PaperText | FetaTab | Avg |
| MDocAgent$_t$ | ✓ | ✗ | ✓ | 0.287 | 0.508 | 0.196 | 0.376 | 0.552 | 0.384 |
| MDocAgent$_t$ | ✓ | ✓ | ✗ | 0.288 | 0.484 | 0.201 | 0.391 | 0.596 | 0.392 |
| MDocAgent$_s$ | ✗ | ✓ | ✓ | 0.285 | 0.479 | 0.188 | 0.365 | 0.592 | 0.382 |
| **MDocAgent** | ✓ | ✓ | ✓ | **0.299** | **0.517** | **0.219** | **0.399** | **0.600** | **0.407** |

Table 3: Performance comparison across different evidence source on MMLongBench.

| Method | Chart | Table | Pure-text | Generalized-text | Figure | Avg |
|---|---|---|---|---|---|---|
| *LVLMs (up to 32 pages)* | | | | | | |
| Qwen2-VL-7B-Instruct | 0.182 | 0.097 | 0.209 | 0.185 | 0.197 | 0.165 |
| Qwen2.5-VL-7B-Instruct | 0.188 | 0.124 | 0.265 | 0.210 | 0.254 | 0.224 |
| LLaVA-v1.6-Mistral-7B | 0.011 | 0.023 | 0.033 | 0.000 | 0.057 | 0.074 |
| LLaVA-One-Vision-7B | 0.045 | 0.051 | 0.076 | 0.017 | 0.084 | 0.053 |
| Phi-3.5-Vision-Instruct | 0.159 | 0.101 | 0.156 | 0.160 | 0.164 | 0.144 |
| SmolVLM-Instruct | 0.062 | 0.065 | 0.123 | 0.118 | 0.094 | 0.081 |
| *RAG methods (top 1)* | | | | | | |
| ColBERTv2+LLaMA-3.1-8B | 0.148 | 0.203 | 0.265 | 0.143 | 0.074 | 0.241 |
| M3DocRAG (ColPali+Qwen2-VL-7B) | 0.268 | 0.263 | 0.334 | 0.250 | **0.303** | 0.276 |
| **MDocAgent (Ours)** | **0.269** | **0.300** | **0.348** | **0.252** | 0.298 | **0.299** |
| *RAG methods (top 4)* | | | | | | |
| ColBERTv2+LLaMA-3.1-8B | 0.182 | 0.267 | 0.311 | 0.168 | 0.120 | 0.273 |
| M3DocRAG (ColPali+Qwen2-VL-7B) | 0.290 | 0.318 | 0.371 | 0.277 | **0.321** | 0.296 |
| **MDocAgent (Ours)** | **0.347** | **0.323** | **0.401** | **0.294** | 0.321 | **0.315** |

**MDocAgent Outperforms All the Comparison Methods and Other LVLMs**. We compare our method with baseline approaches on document understanding tasks, with the results presented in Table 1. Overall, our method outperforms all baselines across all benchmarks.

**Top-1 Retrieval Performance.** With top-1 retrieval, MDocAgent demonstrates a significant performance improvement. On PaperText, MDocAgent achieves a score of 0.399, surpassing the second-best method, M3DocRAG, by 16.7%. Similarly, on FetaTab, MDocAgent attains a score of 0.600, exceeding the second-best method by an impressive 21.0%. Compared to the best LVLM (Qwen2.5-VL-7B) and text-RAG-based (ColBERTv2+Llama-3.1-8B) baselines, our approach demonstrates a remarkable average improvement of 51.9% and 23.7% on average across all benchmarks. This improvement highlights the benefits of incorporating visual information and the collaborative multi-agent architecture in our framework. Furthermore, recent state-of-the-art image-RAG-based method M3DocRAG (Cho et al., 2024) show promising results, yet our approach still outperforms it by 12.1% on average. This suggests that our multi-agent framework, with its specialized agents and critical information extraction mechanism addresses the core challenges of information overload, granular attention to detail, and cross-modality understanding more effectively than existing methods.

**Top-4 Retrieval Performance.** When using top-4 retrieval, the advantages of our method are further demonstrated. MDocAgent consistently achieves the highest scores across all benchmarks. On average, MDocAgent outperforms Qwen2.5-VL-7B by a remarkable 73.5%. Interestingly, with top-4 retrieval, M3DocRAG slightly performs worse than ColBERTv2+Llama-3.1-8B compared to top-1 retrieval. This may suggest limitations on M3DocRAG's capacity of selectively integrate across multiple retrieved documents when dealing with larger amounts of retrieved information. On average, MDocAgent exceeds M3DocRAG by 10.9%. Meanwhile, compared to ColBERTv2+Llama-3.1-8B, MDocAgent demonstrates a 6.9% improvement. This consistent improvement suggests that our method effectively harnesses the additional contextual information provided by the top-4 retrieved items, offering a greater benefit with more retrieval results.

Table 4: Performance comparison between using ColPali and ColQwen2-v1.0 as MDocAgent's image-based RAG model.

| | MMLongBench | LongDocUrl | PaperTab | PaperText | FetaTab | Avg |
|---|---|---|---|---|---|---|
| **+ColPali** | 0.299 | 0.517 | **0.219** | **0.399** | 0.600 | **0.407** |
| **+ColQwen2-v1.0** | **0.303** | **0.520** | 0.216 | 0.391 | **0.603** | **0.407** |

## 4.3 QUANTITATIVE ANALYSIS

In this section, we conduct three quantitative analyses to understand the effectiveness and contribution of different components within our proposed framework. First, we perform ablation studies to assess the impact of removing individual agents or groups of agents. Second, we present a fine-grained performance analysis, examining MDocAgent's performance across different evidence modalities on MMLongBench to pinpoint the source of its improvements. Third, a compatibility analysis explores the framework's performance with different image-based RAG backbones to demonstrate its robustness and generalizability. Additionally, we present experimental results showcasing its performance with different model backbones in Appendix B.2.

### 4.3.1 ABLATION STUDIES

Table 2 presents a comparison of our full method (MDocAgent) against it's variants: $MDocAgent_i$ (without the text agent) and $MDocAgent_t$ (without the image agent). Across all benchmarks, the full MDocAgent method consistently achieves the highest performance. The removal of either specialized agent, text or image, results in a noticeable performance drop. This underscores the importance of incorporating both text and image modalities through specialized agents within our framework. The performance difference is most pronounced in benchmarks like LongDocURL and PaperText, which likely contain richer visual or textual information respectively, further highlighting the value of specialized processing. This ablation study clearly demonstrates the synergistic effect of combining specialized agents dedicated to each modality.

Table 2 also compares MDocAgent with $MDocAgent_s$, where both the general agent and the critical agent are removed, to evaluate their contribution. The consistent improvement of the full method over $MDocAgent_s$ across all datasets clearly underscores the importance of these two agents. The general agent establishes a crucial foundation by initially integrating both text and image modalities, providing a holistic understanding of the context. Removing this integration step noticeably reduces the subsequent agents' capacity to focus their analysis of critical information and answer effectively. On top of general modal integration, removing the critical agent limits the framework's ability to effectively identify and leverage crucial information. This highlights the essential role of the critical agent in focusing the specialized agents' attention and facilitating more targeted and efficient information extraction.

### 4.3.2 FINE-GRAINED PERFORMANCE ANALYSIS

We present an in-depth analysis of the performance in different types of evidence modalities, by further analyzing the scores on MMLongBench in Table 3, to gain a better understanding of the performance improvements achieved by MDocAgent. We also illustrate the results of evidence modalities of LongDocURL in Appendix B.1. According to the results, MDocAgent outperforms all LVLM baselines among all types of evidence modalities. When comparing RAG methods using the top 1 retrieval approach, though M3DocRAG performs slightly better on Figure category, MDocAgent show strong performance in Chart, Table and Text categories, reflecting its enhanced capability to process textual and visual information. With the top 4 retrieval strategy, MDocAgent enhances its performance in the all categories, specifically in Figure, highlighting its effective handling of large and varied information sources. We also provide experiments and analysis about different document lengths and retrieval module performance in Appendix B.3 and B.4.

### 4.3.3 COMPATIBILITY ANALYSIS

We further analyze the compatibility of MDocAgent with different RAG backbones. Table 4 presents results using two image-based RAG models, ColPali and ColQwen2-v1.0, within our proposed

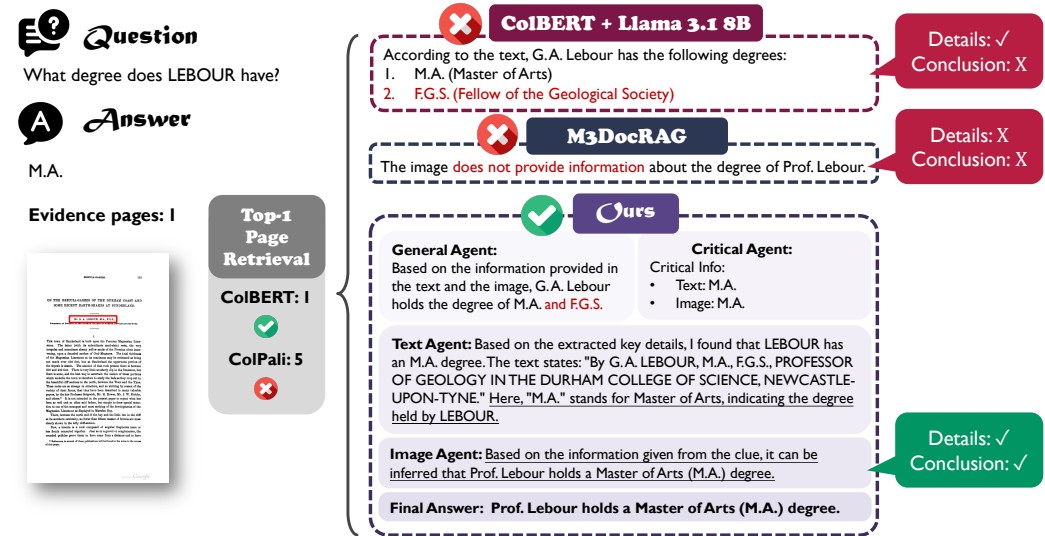

Figure 3: A Case study of **MDocAgent** compared with other two RAG-method baselines. In this case, ColPali fails to retrieve the correct evidence page, hindering M3DocRAG. While ColBERT succeeds in retrieval, the ColBERT + Llama baseline still provides an incorrect answer. Only our multi-agent framework, through precise critical information extraction and agent collaboration, correctly identifies the M.A. degree.

framework. Both models achieve comparable overall performance, with an identical average score of 0.407 across all benchmarks. While ColQwen2-v1.0 shows a slight advantage on MMLongBench, LongDocUrl, and FetaTab, ColPali performs marginally better on PaperTab and PaperText. This suggests that the choice of image-based RAG model has minimal impact on the framework's overall effectiveness, underscoring the robustness of our multi-agent architecture. Moreover, the consistency in performance across different RAG models highlights that the core strength of our approach lies in the multi-agent architecture itself, rather than reliance on a specific retrieval model. This further reinforces the compatibility of our proposed method.

## 4.4 CASE STUDY

We perform a case study to better understand MDocAgent. Figure 3 illustrates an example. The question asks for Professor Lebour's degree. ColPali fails to retrieve the relevant page, rendering M3DocRAG ineffective. While ColBERT correctly retrieves the page, ColBERT + Llama still produces an incorrect answer because it incorrectly adds "F.G.S." to the answer, which is not a degree. MDocAgent, on the other hand, correctly identifies the "M.A. degree". The general agent provides an initial answer, and the critical agent identifies the "M.A." designation in both text and image. Based on the clue, the text agent adds a more detailed explanation, and the image agent directly uses the clue as its answer. Finally, the summarizing agent synthesizes the results to provide the verified answer. This case study demonstrates how our structured, multi-agent framework outperforms methods struggling with integrated text and image analysis (See more case studies in Appendix B.5).

## 5 CONCLUSION

This paper presents a multi-agent framework MDocAgent for DocQA that integrates text and visual information through specialized agents and a dual RAG approach. Our framework addresses the limitations of existing methods by employing agents dedicated to text processing, image analysis, and critical information extraction, culminating in a synthesizing agent for final answer generation. Experimental results demonstrate significant improvements over LVLMs and multi-modal RAG methods, highlighting the efficacy of our collaborative multi-agent architecture. Our framework effectively handles information overload and promotes detailed cross-modal understanding, leading to more accurate and comprehensive answers in complex DocQA tasks. Future work will explore more advanced inter-agent communication and the integration of external knowledge sources.

## 6 ETHICS STATEMENT

This work does not involve human subjects, personal data collection, or any sensitive demographic attributes. All datasets used in our experiments (MMLongBench, LongDocURL, PaperTab, PaperText, and FetaTab) are publicly available, and we strictly follow their respective licenses and intended usage. We have carefully considered potential risks: our framework improves document question answering by integrating textual and visual cues, but does not generate or manipulate personal or confidential data. To mitigate concerns of bias or fairness, we report results across multiple benchmarks covering diverse modalities and domains. We further note that no funding sources or sponsorships introduce conflicts of interest. Overall, the research complies with standard practices of research integrity, reproducibility, and legal compliance.

## 7 REPRODUCIBILITY STATEMENT

We have made extensive efforts to ensure the reproducibility of our results. All experimental settings, including datasets, models, and hyperparameters, are described in detail in Section 4 and Appendix A. We provide ablation studies and compatibility analyses (Sections 4.3 and B.2) to clarify the contributions of different components. Complete descriptions of evaluation metrics and prompts are provided in Appendix A.4. In addition, we indicate the usage of open-source retrieval models (ColBERTv2, ColPali) and base LLMs (Llama-3.1, Qwen2-VL, GPT-4o). Anonymous code and implementation scripts, including preprocessing and evaluation, are submitted as supplementary material. These resources collectively allow independent researchers to replicate and validate our findings.

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

# A  EXPERIMENTAL SETUP

## A.1  BASELINE MODELS

- **Qwen2-VL-7B-Instruct** (Wang et al., 2024): A large vision-language model developed by Alibaba, designed to handle multiple images as input.

- **Qwen2.5-VL-7B-Instruct** (Bai et al., 2025): An enhanced version of Qwen2-VL-7B-Instruct, offering improved performance in processing multiple images.

- **llava-v1.6-mistral-7b** (Liu et al., 2024a): Also called LLaVA-NeXT, a vision-language model improved upon LLaVa-1.5, capable of interpreting and generating content from multiple images.

- **Phi-3.5-vision-instruct** (Abdin et al., 2024): A model developed by Microsoft that integrates vision and language understanding, designed to process and generate responses based on multiple images.

- **llava-one-vision-7B** (Li et al., 2024): A model trained on LLaVA-OneVision, based on Qwen2-7B language model with a context window of 32K tokens.

- **SmolVLM-Instruct** (Marafioti et al., 2025): A compact vision-language model developed by HuggingFace, optimized for handling image inputs efficiently.

- **ColBERTv2+Llama-3.1-8B-Instruct** (Santhanam et al., 2021; Grattafiori et al., 2024): A text-based RAG pipeline that utilizes ColBERTv2 (Santhanam et al., 2021) for retrieving text segments and Llama-3.1-8B-Instruct as the LLM to generate responses.

- **M3DocRAG** (Cho et al., 2024): An image-based RAG pipeline that employs ColPali (Faysse et al., 2024a) for retrieving image segments and Qwen2-VL-7B-Instruct (Wang et al., 2024) as the LVLM for answer generation.

Table 5: Performance comparison across different evidence source on LongDocURL.

| Method | Layout | Text | Figure | Table | Others | Avg |
|---|---|---|---|---|---|---|
| *LVLMs* | | | | | | |
| Qwen2-VL-7B-Instruct | 0.264 | 0.386 | 0.308 | 0.207 | 0.500 | 0.296 |
| Qwen2.5-VL-7B-Instruct | 0.357 | 0.479 | 0.442 | 0.299 | 0.375 | 0.389 |
| llava-v1.6-mistral-7b | 0.067 | 0.165 | 0.088 | 0.051 | 0.250 | 0.099 |
| llava-one-vision-7B | 0.098 | 0.200 | 0.144 | 0.057 | 0.125 | 0.126 |
| Phi-3.5-vision-instruct | 0.245 | 0.375 | 0.291 | 0.187 | 0.375 | 0.280 |
| SmolVLM-Instruct | 0.128 | 0.224 | 0.164 | 0.100 | 0.250 | 0.163 |
| *RAG methods (top 1)* | | | | | | |
| ColBERTv2+Llama-3.1-8B | 0.257 | 0.529 | 0.471 | 0.428 | **0.775** | 0.429 |
| M3DocRAG (ColPali+Qwen2-VL-7B) | 0.340 | 0.605 | **0.546** | 0.520 | 0.625 | 0.506 |
| **MDocAgent (Ours)** | **0.341** | **0.612** | 0.540 | **0.527** | 0.750 | **0.517** |
| *RAG methods (top 4)* | | | | | | |
| ColBERTv2+Llama-3.1-8B | 0.349 | 0.599 | 0.491 | 0.485 | **0.875** | 0.491 |
| M3DocRAG (ColPali+Qwen2-VL-7B) | 0.426 | 0.660 | 0.595 | 0.542 | 0.625 | 0.554 |
| **MDocAgent (Ours)** | **0.438** | **0.675** | 0.592 | **0.581** | **0.875** | **0.578** |

## A.2  EVALUATION BENCHMARKS

- **MMLongBench** (Ma et al., 2024b): Evaluates models' ability to understand long documents with rich layouts and multi-modal components, comprising 1091 questions and 135 documents averaging 47.5 pages each.

- **LongDocURL** (Deng et al., 2024): Provides a comprehensive multi-modal long document benchmark integrating understanding, reasoning, and locating tasks, covering over 33,000 pages of documents and 2,325 question-answer pairs.

- **PaperTab** (Hui et al., 2024): Focuses on evaluating models' ability to comprehend and extract information from tables within NLP research papers, covering 393 questions among 307 documents.

- **PaperText** (Hui et al., 2024): Assesses models' proficiency in understanding the textual content of NLP research papers, covering 2804 questions among 1087 documents.

- **FetaTab** (Hui et al., 2024): a question-answering dataset for tables from Wikipedia pages, challengeing models to generate free-form text answers, comprising 1023 questions and 878 documents.

## A.3 HYPERPARAMETER SETTINGS

- **Temperature**: All models use their default temperature setting.

- **Max New Tokens**: 256.

- **Max Tokens per Image (Qwen2-VL-7B-Instruct)**:

    - **Top-1 retrieval**: 16,384 (by default).
    - **Top-4 retrieval**: 2,048.

- **Image Resolution**: 144 (for all benchmarks).

## A.4 PROMPT SETTINGS

---

**General Agent**

You are an advanced agent capable of analyzing both text and images. Your task is to use both the textual and visual information provided to answer the user's question accurately.
**Extract Text from Both Sources**: If the image contains text, extract it and consider both the text in the image and the provided textual content.
**Analyze Visual and Textual Information**: Combine details from both the image (e.g., objects, scenes, or patterns) and the text to build a comprehensive understanding of the content.
**Provide a Combined Answer**: Use the relevant details from both the image and the text to provide a clear, accurate, and context-aware response to the user's question.
**When responding:**

- If both the image and text contain similar or overlapping information, cross-check and use both to ensure consistency.

- If the image contains information not present in the text, include it in your response if it is relevant to the question.

- If the text and image offer conflicting details, explain the discrepancies and clarify the most reliable source.

---

**Critical Agent**

Provide a Python dictionary of critical information based on all given information—one for text and one for image.
Respond exclusively in a valid dictionary format without any additional text. The format should be:
{"text": "critical information for text", "image": "critical information for image"}

---

**Text Agent**

You are a text analysis agent. Your job is to extract key information from the text and use it to answer the user's question accurately.

**Your tasks:**

- Extract key details. Focus on the most important facts, data, or ideas related to the question.
- Understand the context and pay attention to the meaning and details.
- Use the extracted information to give a concise and relevant response to the user's question. Provide a clear answer.

**Image Agent**

You are an advanced image processing agent specialized in analyzing and extracting information from images. The images may include document screenshots, illustrations, or photographs.

**Your tasks:**

- Extract textual information from images using Optical Character Recognition (OCR).
- Analyze visual content to identify relevant details (e.g., objects, patterns, scenes).
- Combine textual and visual information to provide an accurate and context-aware answer to the user's question.

**Summarizing Agent**

You are tasked with summarizing and evaluating the collective responses provided by multiple agents. You have access to the following information:

- **Answers**: The individual answers from all agents.

**Your tasks:**

- **Analyze**: Evaluate the quality, consistency, and relevance of each answer. Identify commonalities, discrepancies, or gaps in reasoning.
- **Synthesize**: Summarize the most accurate and reliable information based on the evidence provided by the agents and their discussions.
- **Conclude**: Provide a final, well-reasoned answer to the question or task. Your conclusion should reflect the consensus (if one exists) or the most credible and well-supported answer.

Return the final answer in the following dictionary format:
{"Answer": Your final answer here}

**Evaluation**

**Question**: {question}
**Predicted Answer**: {answer}
**Ground Truth Answer**: {gt}
Please evaluate whether the predicted answer is correct.

- If the answer is correct, return 1.
- If the answer is incorrect, return 0.

Return only a string formatted as a valid JSON dictionary that can be parsed using `json.loads`, for example: {"correctness": 1}

## A.5 Evaluation Metrics

The metric of all benchmarks is the average binary correctness evaluated by GPT-4o. The evaluation prompt is given in Section A.4. We use a python script to extract the result provided by GPT-4o.

# B Additional Results

## B.1 Fine-grained Performance of LongDocURL

We present the fine-grained performance of LongDocURL, as illustrated in Table 5. Similar to MMLongBench, MDocAgent outperforms all LVLM baselines. When using the top 1 retrieval approach, though M3DocRAG performs slightly better on Figure and ColBERTv2+Llama3.1-8B performs slightly better on the type Others, MDocAgent show strong performance in Layout, Text, Table and get the highest average accuracy. With the top 4 retrieval strategy, MDocAgent improves its performance and reach the highest score in the all categories.

## B.2 Experiments on different model backbones in MDocAgent

Table 6: Performance comparison of using different backbone LVLMs in MDocAgent.

| | MMLongBench | LongDocUrl | PaperTab | PaperText | FetaTab | Avg |
|---|---|---|---|---|---|---|
| *With top 1 retrieval* | | | | | | |
| **+Qwen2-VL-7B-Instruct** | 0.299 | 0.517 | 0.219 | 0.399 | 0.600 | 0.407 |
| **+Qwen2.5-VL-7B-Instruct** | 0.351 | 0.519 | 0.211 | 0.382 | 0.589 | 0.410 |
| **+GPT-4o OpenAI (2023)** | **0.420** | **0.595** | **0.293** | **0.474** | **0.716** | **0.500** |
| *With top 4 retrieval* | | | | | | |
| **+Qwen2-VL-7B-Instruct** | 0.315 | **0.578** | **0.278** | **0.487** | **0.675** | 0.467 |
| **+Qwen2.5-VL-7B-Instruct** | **0.389** | 0.566 | 0.277 | 0.454 | 0.671 | **0.471** |

Table 6 presents an ablation study evaluating the impact of different LVLMs on the performance of our framework. Three LVLMs: Qwen2-VL-7B-Instruct, Qwen2.5-VL-7B-Instruct, and GPT-4o were integrated as the backbone model for all agents except the text agent.

Qwen2.5-VL-7B-Instruct performs worse than Qwen2-VL-7B-Instruct on PaperTab, PaperText, and FetaTab, with both top-1 and top-4 retrieval. However, Qwen2.5-VL shows an extremely marked improvement over Qwen2-VL on MMLongBench, resulting higher average scores. MMLongBench's greater reliance on image-based questions might explain Qwen2.5-VL's superior performance on this benchmark, possibly indicating that Qwen2.5-VL is better at handling visual question-answering tasks, but worse at handling textual tasks.

Importantly, GPT-4o significantly outperforms both Qwen2-VL and Qwen2.5-VL across all benchmarks. Remarkably, GPT-4o's top-1 performance surpasses even the top-4 results of both Qwen models in almost all cases. This substantial performance increase strongly suggests that our framework effectively leverages more powerful backbone models, showcasing its adaptability and capacity to benefit from improvements in the underlying LVLMs.

## B.3 Experiments on different document lengths

We categorized the dataset MMLongBench into three groups based on page count. The results(Table 7) demonstrate that our method consistently outperforms baselines across all document length categories.

Table 7: Comparison of Scores Across Page Ranges

| Method | 1–20 Pages | | 20–40 Pages | | >40 Pages | |
|---|---|---|---|---|---|---|
| | Top 1 | Top 4 | Top 1 | Top 4 | Top 1 | Top 4 |
| ColBert+LLama3.1 | 0.298 | 0.306 | 0.216 | 0.262 | 0.224 | 0.263 |
| M3DocRAG | 0.309 | 0.340 | 0.242 | 0.255 | 0.276 | 0.311 |
| **MDocAgent (Ours)** | **0.347** | **0.381** | **0.250** | **0.269** | **0.291** | **0.319** |

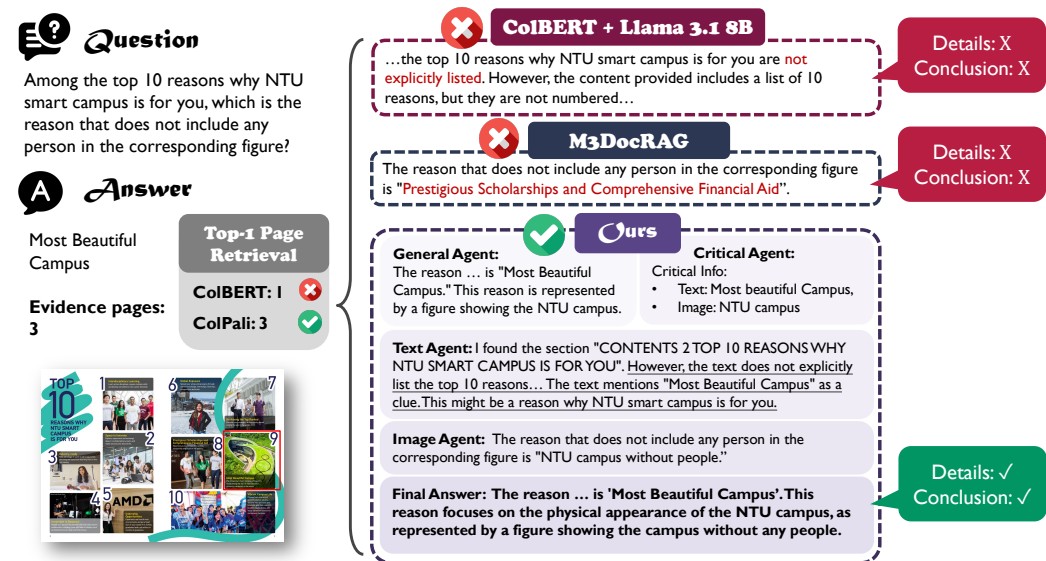

Figure 4: A Case study of **MDocAgent** compared with other two baselines. While only ColPali correctly retrieves the evidence page, neither baseline method identifies the correct answer. Our method, through critical information sharing and specialized agent collaboration, correctly pinpoints the "Most Beautiful Campus" as the only reason without a corresponding image containing people.

## B.4 EXPERIMENTS ON RETRIEVAL MODULE PERFORMANCE

We evaluated our retrieval module on LongDocURL and MMLongBench, analyzing text and image retrieval separately and jointly. We measured the percentage of instances where at least one evidence page was present within the top-k retrieved pages. Results (Table 8) indicate significant accuracy improvements when leveraging both modalities together, demonstrating mutual compensation between text and image retrieval.

Table 8: Comparison of Retrieval Accuracy

| RAG Method | LondDocURL | | MMLongBench | |
|---|---|---|---|---|
| | Top 1 | Top 4 | Top 1 | Top 4 |
| Text | 75.74% | 89.46% | 6.99% | 23.48% |
| Image | 80.77% | 94.84% | 6.99% | 31.33% |
| Mixed Modalities | **88.22%** | **97.08%** | **11.47%** | **38.86%** |

## B.5 ADDITIONAL CASE STUDIES

In Figure 4, the question requires identifying a reason from a list that lacks explicit numbering and is accompanied by images. ColBERT fails to retrieve the correct evidence page, resulting ColBERT + Llama's inability to answer the question. Although ColPali correctly locates the evidence page, M3DocRAG fails to get the correct answer. However, our framework successfully identifies the correct answer ("Most Beautiful Campus") through the concerted efforts of all agents. The general agent arrives at a preliminary answer and the critical agent identifies critical textual clues ("Most Beautiful Campus") and corresponding visual elements (images of the NTU campus). Image agent then refines the answer, leveraging the critical information to correctly pinpoint the description lacking people. Though text agent can't find the related information from the given context, information provided by the critical agent helps it to guess that the answer is "Most Beautiful Campus". The summarizing agent combines these insights to arrive at the correct final answer.

In Figure 5, the question requires extracting and comparing numerical information related to two distinct Latino populations from both textual and tabular data within a document. While both ColBERT and ColPali successfully retrieve the relevant page containing the necessary information,

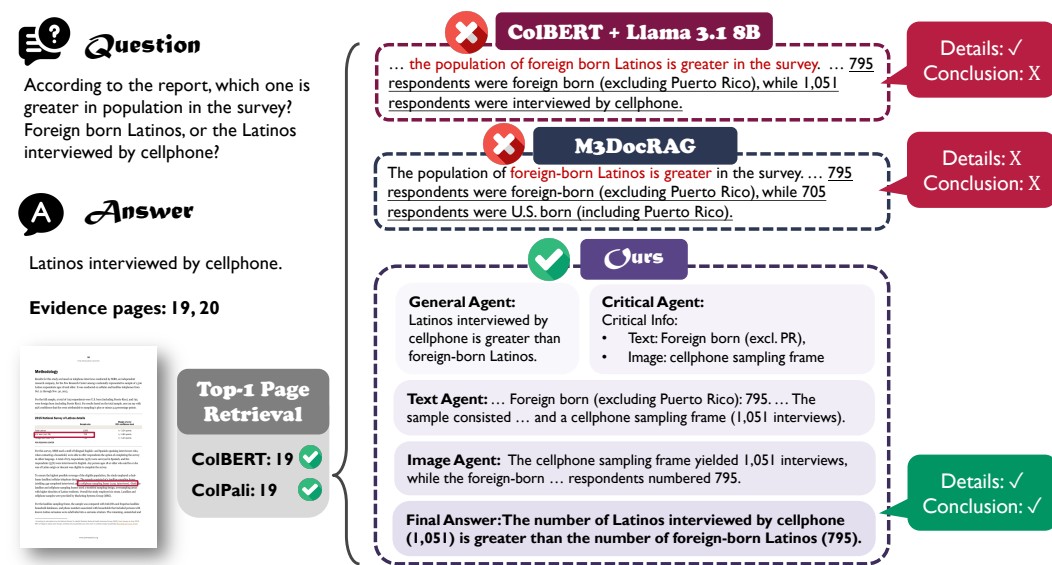

Figure 5: A Case study of **MDocAgent** compared with other two baselines. Although correctly retrieve the results, both baseline methods fail to arrive at the correct answer. Our framework, through the collaborative efforts of its specialized agents, successfully identifies the relevant information from both text and a table within the image, ultimately synthesizing the correct answer.

both baseline methods fail to synthesize the correct answer. The ColBERT + Llama-3.1-8B baseline, relying solely on text, incorrectly concludes that the foreign-born Latino population is greater, demonstrating a failure to accurately interpret the numerical data presented within the document's textual content. Similarly, M3DocRAG fails to correctly interpret the question due to capturing wrong information. In contrast, our multi-agent framework successfully navigates this complexity and gives the correct answer. Specifically, the general agent provides a correct but vague answer, making the critical agent essential for identifying key phrases like "Foreign born (excl. PR)" and the "cellphone sampling frame" table. This guides specialized agents to precise locations for efficient data extraction. Both text agent and image agent correctly extract 795 for foreign-born Latinos and 1,051 for cellphone-interviewed Latinos. The summarizing agent then integrates these insights for accurate comparison and a comprehensive final answer.

These two cases highlight MDocAgent's resilience to imperfect retrieval, demonstrating the effectiveness of collaborative multi-modal information processing and the importance of the general-critical agent's guidance in achieving high accuracy even with potentially insufficient or ambiguous information.

## C DISCLOSURE OF LARGE LANGUAGE MODEL USAGE

All content in this article is entirely authored by the writers. The LLM was used solely for language refinement and stylistic polishing, without contributing to content generation. All LLM-refined passages were subsequently reviewed and revised by the authors.

