# OpenReview forum: "MDocAgent: A Multi-Modal Multi-Agent Framework for Document Question Answering"
_ICLR.cc/2026/Conference — Submitted to ICLR 2026_

### Official Review · Reviewer_J2E6 · 2025-10-20

**Soundness:** 2
**Presentation:** 3
**Contribution:** 2
**Rating:** 4
**Confidence:** 5

**Summary:**

To address DocQA limitations (single-modality bias, isolated RAG, long-document overload), this paper proposes MDocAgent—a framework integrating dual RAG (text via ColBERTv2, image via ColPali) and 5 collaborative agents (General, Critical, Text, Image, Summarizing). Evaluated on 5 benchmarks (MMLongBench, FetaTab, etc.), it outperforms baselines: Top-1 accuracy 0.407 (new SOTA), Top-4 0.465. Ablation confirms all agents are necessary. Key contributions: "dual RAG + multi-agent" architecture, critical info extraction to reduce agent attention dispersion, and validation for complex multi-modal docs.

**Strengths:**

1. **Originality**: First to combine dual-modal RAG with specialized multi-agents for DocQA; Critical Agent’s info-filtering solves agent attention dispersion (a unique fix for multi-agent noise).
2. **Quality**: Rigorous experiments (5 diverse benchmarks, comprehensive baselines) and ablation studies (proving each agent’s value) reduce bias.
3. **Clarity**: Logical flow (background→design→experiments) + consistent terminology (e.g., $T_q$/$I_q$ for retrieval) and visualizations aid understanding.

**Weaknesses:**

1. **No computational efficiency metrics**: Dual RAG + 5 agents may be costly, but no inference time/memory data vs. lightweight baselines (e.g., Qwen2-VL-7B). *Fix*: Add efficiency tests (A100 latency, memory) and optimize (e.g., RAG caching) if needed.
2. **Limited non-standard doc testing**: Only uses academic/structured docs—unproven on noisy/scanned docs (e.g., medical records, historical texts). *Fix*: Small-scale tests on 1–2 non-standard datasets (e.g., PubMed Central) to check adaptability.
3. **Lack of agent base model replacement tests in ablations**: Current ablations only remove entire agents, not replacing their base models with alternatives. This fails to clarify if performance gains stem from the **multi-agent architecture itself** or just the "strong-strong combination" of high-performance base models for agents.
4. **Insufficient novelty due to heavy reliance on existing component stacking**: The work largely assembles pre-existing technologies (ColBERTv2/ColPali for RAG, standard LLMs/LVLMs for agent backbones) without introducing core innovative mechanisms beyond "combining dual RAG with multi-agents." This makes the framework feel more like a "component stack" than a breakthrough in multi-modal DocQA.

**Questions:**

1. Have you tested MDocAgent with simpler RAG backbones (e.g., BM25 for text, CLIP for images) to see if performance depends on high-quality retrieval?

---

### Official Review · Reviewer_TnuR · 2025-10-26

**Soundness:** 3
**Presentation:** 3
**Contribution:** 2
**Rating:** 2
**Confidence:** 4

**Summary:**

This paper introduces MDocAgent, a multi-modal multi-agent framework for document question answering (DocQA). Unlike traditional LLM-based or LVLM-based RAG systems that typically focus on a single modality (text or image), MDocAgent integrates both textual and visual information through five collaborative agents.
The system leverages dual RAG pipelines (ColBERTv2 for text and ColPali for images) to retrieve the most relevant segments and pages, and then coordinates these agents through staged reasoning and synthesis.
Experiments across five benchmarks (MMLongBench, LongDocURL, PaperTab, PaperText, and FetaTab) show an average improvement of 12.1% over current state-of-the-art RAG methods (like M3DocRAG).

**Strengths:**

1. The proposed framework demonstrates clear and consistent performance gains (both in top-1 and top-4 retrieval) across multiple datasets, showing robustness and generalizability.
2. The five-stage design and algorithmic flow are clearly explained, offering a transparent view of agent interactions.

**Weaknesses:**

1. While the multi-agent integration is interesting, each component (RAG, multi-modal LVLMs, and multi-agent orchestration) is individually well-known. The main contribution lies in combining them, which may be viewed as engineering rather than fundamental innovation.
2. Agents appear to operate in a mostly sequential pipeline without explicit feedback loops or collaborative reasoning (e.g., debate, negotiation, or reflection). This weakens the “multi-agent” claim. I would prefer calling it pipeline for this approach.
3. Only one case study is presented. A systematic error analysis would help illustrate remaining weaknesses (e.g., conflicting textual and visual cues, long-context compression).

**Questions:**

Were any of the agents fine-tuned for their specific subtask, or were they purely zero-shot with instruction prompts?
How were the prompt templates for each agent optimized (manual design vs. automatic tuning)?

---

### Official Review · Reviewer_k5Qh · 2025-10-29

**Soundness:** 2
**Presentation:** 3
**Contribution:** 2
**Rating:** 4
**Confidence:** 4

**Summary:**

This paper presents MDocAgent, a multi-modal, multi-agent framework for document understanding and question answering. The system integrates both text- and image-based retrieval (via ColBERT and ColPali) and coordinates several specialized agents (text, image, critical, and summarizing) to perform collaborative reasoning over multimodal documents. Experimental results on multiple DocQA benchmarks show consistent improvements over existing baselines.

**Strengths:**

Clear motivation
The paper addresses an important challenge in multimodal document understanding, namely how to effectively integrate visual and textual information in long, complex documents.

Well-written and structured
The paper is easy to follow, and the modular multi-agent design is clearly explained.

Empirical gains
The model demonstrates measurable improvements on several benchmarks, indicating engineering effort and sound experimental execution.

**Weaknesses:**

Limited novelty beyond existing frameworks.
The proposed architecture largely extends existing RAG pipelines and multi-agent prompting frameworks without introducing a fundamentally new mechanism or learning paradigm.

The use of multiple specialized agents (e.g., text agent, image agent, summarizing agent) is conceptually similar to prior multi-agent LLM systems such as CAMEL, AutoGen, and MetaGPT, where different roles are assigned to submodels with predefined instructions.

The multi-modal retrieval setup (ColBERT for text, ColPali for image) and subsequent fusion is reminiscent of V-RAG (CVPR 2025), which already demonstrated joint text–image retrieval and multimodal reasoning on large-scale document collections.
Hence, the contribution feels more incremental — essentially combining known RAG + multi-agent ideas rather than proposing a new algorithmic insight.

No clear learning innovation.
The framework relies entirely on prompting and modular coordination, with no evidence of new learning objectives, optimization schemes, or model-level training innovations. As such, the work seems more of a system-level orchestration rather than a methodological advancement, which may fall short of ICLR’s bar for originality.

Unclear attribution of gains.
While performance improvements are reported, it is not clear which component (multi-agent coordination vs. multimodal retrieval) contributes most to the gains. There is no strong ablation isolating each agent’s impact, making it difficult to judge whether the multi-agent design itself truly improves reasoning or simply adds redundancy.

Overlap with DocHaystack / V-RAG.
The paper’s multimodal retrieval backbone and evaluation setup appear to heavily rely on previously published components (e.g., CLIP/SigLIP-based vision retrievers, document-level QA benchmarks). Without major conceptual differences, the contribution risks being viewed as a replication with added agents rather than a new research direction.

Lack of theoretical or analytical insight.
The work does not offer new theoretical understanding or analysis about why or when multi-agent coordination helps multimodal reasoning. This limits the paper’s broader scientific contribution beyond empirical gains.

**Questions:**

Provide deeper analysis of agent collaboration dynamics — e.g., through attention visualization, message-passing traces, or error decomposition — to show how multi-agent reasoning yields more accurate results.

Introduce a new coordination or learning mechanism (e.g., adaptive role assignment, reinforcement-based agent reward, or differentiable communication channel) to strengthen novelty.

Consider benchmarking against multi-agent LLM baselines (e.g., AutoGen, ChatArena) to better contextualize the system-level improvement.

Expand the ablation studies to measure contributions from each agent, and provide evidence that multi-agent synergy, not only multi-modal retrieval, drives the observed performance.

---

### Official Review · Reviewer_f24H · 2025-10-30

**Soundness:** 2
**Presentation:** 3
**Contribution:** 2
**Rating:** 4
**Confidence:** 5

**Summary:**

This paper proposes a multi-agent RAG framework to enhance document VQA. The motivation to integrate multimodal information for RAG-based document understanding is clear and relevant. The authors explore using multiple retrievers combined with different prompting strategies to progressively integrate information and improve performance. While the experimental results demonstrate potential, the paper’s novelty is limited. The approach mainly relies on prompt-based fusion of retrieval results from different modalities without introducing substantial methodological innovation. Moreover, as a training-free framework, the experimental validation remains limited.

**Strengths:**

- The motivation of this paper is meaningful, which points out the limitations in this domain, especially for long document understanding, and the paper's structure and writing are easy to follow.
- The method looks potentially technically and practically workable and has been evaluated on various benchmark datasets.

**Weaknesses:**

- Novelty concerns: While the proposed framework is practical and the workflow appears effective, the overall novelty is limited. The system mainly relies on existing trained or off-the-shelf retrievers, with information fusion achieved through different prompting methods. Further improvements are expected in retrieval quality, refinement, and cross-page entity correlation understanding.

- Evaluation limitations: The evaluation depends solely on LLM-based metrics assessed by GPT-4o, which is insufficient. Additionally, the model is tested on a single LLM, lacking evidence of robustness across different model types (e.g., varying sizes, open- vs. closed-source).

- Qualitative analysis: The paper presents mainly correct cases, whereas including error cases would provide a more comprehensive analysis.

- Reasoning analysis: Although the paper frequently emphasizes the framework’s “reasoning” ability, it does not clearly demonstrate how reasoning is enhanced or which aspects of reasoning are improved.

**Questions:**

- Handling missing retrievals: The paper does not clarify how the framework addresses cases where the initial retrievers fail to capture critical content. A mechanism for iterative retrieval, fallback strategies, or retrieval refinement would strengthen the design.


- Comparison with closed-source models: It is unclear whether the proposed approach has been compared against closed-source baselines (e.g., GPT-4-Turbo, Claude, Gemini). Such comparisons would help contextualize performance and demonstrate competitiveness.


- Retriever adaptation: The paper does not mention whether any pretraining or domain-specific fine-tuning was applied to the retrievers. Details on retrieval performance and domain adaptation are needed to assess the robustness and generalization of the approach.

---

### Meta-Review · Area_Chair_NCZQ · 2026-01-01

**Summary:**

DocQA is a common challenge. However, existing LLM/LVLM + RAG approaches tend to prioritize single modalities. This results in an insufficient integration of text and visual cues. This hinders performance gains in complex multimodal reasoning. Therefore, this paper proposes MDocAgent, a RAG + multi-agent framework that utilizes text and images. It demonstrates that five agents perform multimodal retrieval. These agents are General, Critical, Text, Image, and Summarization. They integrate the obtained insights. This improves document comprehension and QA accuracy.

However, reviewers raised numerous concerns. For example, the novelty is weak because it focuses primarily on existing retrievers and prompting. The evaluation relies heavily on GPT-4o and lacks sufficient generalization testing. MDocAgent appears to be a patchwork of existing RAG + multi-agent approaches, lacking any learning innovation or theoretical insight. Systematic error analysis is insufficient. Computational efficiency (time/memory) is not evaluated.

The authors did not submit rebuttals addressing these concerns, leaving them unresolved. Therefore, the AC has decided to reject this paper.

**Reviewer Concerns:**

Because the authors did not submit a rebuttal, all concerns remain unresolved.

**Reviewer Scores:**

This paper received negative evaluations from all reviewers. Since the authors did not submit a rebuttal addressing the reviewers’ concerns, the scores did not change.

---

### Decision · Program_Chairs · 2026-01-26

Reject